# Long-Period Fiber Grating Sensors for Chemical and Biomedical Applications

**DOI:** 10.3390/s23010542

**Published:** 2023-01-03

**Authors:** Jintao Cai, Yulei Liu, Xuewen Shu

**Affiliations:** Wuhan National Laboratory for Optoelectronics & School of Optical and Electronic Information, Huazhong University of Science and Technology, Wuhan 430074, China

**Keywords:** biosensors, optical fiber sensor, long-period fiber gratings, sensitivity enhancement, functionalization method

## Abstract

Optical fiber biosensors (OFBS) are being increasingly proposed due to their intrinsic advantages over conventional sensors, including their compactness, potential remote control and immunity to electromagnetic interference. This review systematically introduces the advances of OFBS based on long-period fiber gratings (LPFGs) for chemical and biomedical applications from the perspective of design and functionalization. The sensitivity of such a sensor can be enhanced by designing the device working at or near the dispersion turning point, or working around the mode transition, or their combination. In addition, several common functionalization methods are summarized in detail, such as the covalent immobilization of 3-aminopropyltriethoxysilane (APTES) silanization and graphene oxide (GO) functionalization, and the noncovalent immobilization of the layer-by-layer assembly method. Moreover, reflective LPFG-based sensors with different configurations have also been introduced. This work aims to provide a comprehensive understanding of LPFG-based biosensors and to suggest some future directions for exploration.

## 1. Introduction

Biosensors play a crucial role in environmental and human health monitoring through the efficient and accurate detection of chemistry and biomass. A biosensor normally contains two main components: a bioreceptor and a transducer [1]. When designing a biosensor, it is a key technological step to select the sensitive material suitable for the biological target. Taking the properties of the resulting compounds into account, it is another important step to select the transducer according to the chemical or physical changes caused by the molecular interaction between the biological target and the bioreceptor. The production or consumption of light, heat and chemical substances in the process of recognition is converted into measurable signals [2]. According to these mechanisms, the appropriate transducer can be selected. The information generated by recognition processes is diverse; therefore, the sensors based on electrochemistry, thermotics and optics have been extensively studied.

Optical sensors, especially those based on optical fibers, are being increasingly proposed in the chemical and biomedical applications due to their intrinsic advantages over conventional sensors. Optical fiber biosensors (OFBS) are usually compact and small in size [3] and thus can work with minimal sample volumes. Moreover, OFBS are capable of multiplexing, immune to electromagnetic interference [4] and remote-controllable. The number of publications on OFBS has also steadily increased over the last two decades (as shown in Figure 1). The typical working principle of OFBS is based on monitoring the change of the surrounding refractive index (SRI), produced from the biological target binding on the surface of the fiber, which leads to the alteration of the detectable spectrum [5]. In order to enhance their sensitivity to the SRI, the OFBS are usually machined into microstructures and geometrically modified to generate a strong evanescent field in the sensing area. Examples of such devices include U-shaped and tapered optical fibers [6,7,8,9,10], D-shaped optical fibers [11,12], etched or tilted fiber Bragg gratings (FBGs) [13,14,15], long-period fiber gratings (LPFGs) [16,17,18,19,20], lossy mode resonances (LMRs) [21,22], surface plasmon resonance (SPR) [23,24] Mach–Zehnder interferometers (MZIs) [25,26], photonic-crystal fiber (PCF) [27,28] and Lab-on-Fiber devices [29,30,31]. Among these, some geometrically modified fibers must completely or partially remove the cladding, which affects the robustness of optical fibers, which is a property that should be considered in practical applications. Fortunately, an ideal alternative to avoid this damaging operation is refractive index modulation inside the optical fiber; the grating-based fiber maintains a robust structure and high sensitivity, especially LPFGs, which have been widely used in refractive index sensing, including biochemical molecule sensing [32,33], gas sensing [34,35,36] and ion sensing [37,38].

In this review, we focused attention on LPFG-based sensors for chemical and biomedical applications from the perspective of design and functionalization. Firstly, the principle of LPFG-based biosensors is systematically introduced. Then, several methods to improve the sensitivity of such a sensor are summarized and their applications in the field of biochemistry are illustrated. Furthermore, we categorically describe the biosensing applications of such a sensor in detail according to several typical functionalization methods. Finally, we summarize this review and suggest some future directions for exploration.

## 2. Principle of LPFG-Based Biosensors

An LPFG is a periodic modulation (usually with period Λ = 100–1000 µm) of RI in the fiber core, as shown in Figure 2; the modulation can couple light from the fundamental core mode (LP_01_ mode) to forward-propagating cladding modes (LP_0 *m*_ modes, where *m* = 2, 3, 4, …) and, therefore, produces a set of resonant attenuation bands centered at discrete wavelengths in the LPFG transmission spectrum [18]. Additionally, the phase-matching condition between the effective propagation constants of the LP_01_ mode and LP_0 *m*_ modes can be expressed by [39]:(1)$$\beta_{01}-\beta_{cl}^{\left(m\right)}=\frac{2\pi }{\Lambda }$$
where $\beta_{01}$ and $\beta_{cl}^{\left(m\right)}$ are the propagation constants for the LP_01_ mode and LP_0 *m*_ modes, respectively. Λ is the grating period. For the effective RI (*n_eff_*) of the LP_01_ mode and LP_0 *m*_ modes, they can be expressed by $n_{eff,co}=\frac{\beta_{01}\lambda }{2\pi }$ and $n_{eff,clad}^{\left(m\right)}=\frac{\beta_{cl}^{\left(m\right)}\lambda }{2\pi }$, respectively. In this case, the resonant wavelengths (*λ_res_*) are determined by the effective RI of the LP_01_ mode and LP_0 *m*_ modes, and we can obtain the following expression [40]:(2)$$\lambda_{res}=\left(n_{eff,co}-n_{eff,clad}^{\left(m\right)}\right)\;\Lambda$$

For a specific LPFG, Λ can be a constant and $n_{eff,co}$ is determined by the RI of the core and cladding, whereas $n_{eff,clad}^{\left(m\right)}$ depends on the difference between the cladding RI and SRI of the cladding. Therefore, the resonance characteristic is affected by the effective RI of the cladding region, which is the basic principle of biological detection. In this way, a shift in the resonance wavelength can be represented by the SRI changes caused by molecular interaction occurring on the fiber surface. Naturally, an LPFG-based biosensor can be achieved by equipping the sensor with chemically responsive coatings to selectively detect the biological targets.

## 3. RI Sensitivity Enhancement of LPFG-Based Biosensor

For designing an LPFG-based biosensor, the most important work should be focused on how to enhance the sensitivity of such a sensor. Moreover, a detailed investigation shows that the best performance of sensors based on RI sensitivity can be achieved when the SRI is close to the cladding RI (i.e., 1.44–1.46 RIU) [41]. It may be noted that most biosensors are designed to work in aqueous solutions (i.e., 1.33–1.34 RIU), which is very different from the RI of the fiber cladding, resulting in the device exhibiting low sensitivity to aqueous samples. To address this challenge, several approaches have been proposed over these years. The performance of these methods for LPFG-based biosensors is summarized in Table 1.

### 3.1. Dispersion Turning Point

The most popular methodology is to design a period and cladding mode at or near the dispersion turning point (DTP) according its phase-matching curve (PMC), thereby coupling the forward-propagating core mode with a high-order cladding mode [62]. As shown in Figure 3, the relationship between the grating period and resonance wavelength is demonstrated by the PMC, and the DTP can be observed in the PMC for each cladding mode, where the slope for the higher-order mode changes from positive to negative as the wavelength increases. Hence, for a given grating period, two resonant wavelengths on either side of the DTP are coupled to one cladding mode, forming dual-peak resonances. The spectral difference between the dual peaks can be as the measuring parameter, due to its wavelength changing with the response of LPFGs to the SRI change. In our previous work [18], we theoretically verified that the maximum sensitivity of LPFGs to SRI can be obtained at or near the DTP.

LPFG-based sensors designed near their DTP have been widely used in the field of biochemical sensing throughout the years, due to their high sensitivity to aqueous samples. However, it may be noted that the LPFG based on a DTP sensor with a broad band splits into dual-peak resonances when the device is immersed in the surrounding environment with an RI higher than air (i.e., water). In this case, the device may have a low sensitivity factor, for the reason that the resonant wavelength is far from the DTP. To alleviate this problem, the group of P. Biswas has theoretically and experimentally proved that the sensitivity can be enhanced by tailoring the initial coupling strength of the cladding mode to a specific higher-order cladding mode at the DTP [63].

F. Chiavaioli et al. reported an LPFG-based biosensor near its DTP, with a grating period of 165 μm, for the detection of anti-IgG [42]. Figure 4a shows the simulation result of the dependence between the PMC for the LP_0,12_ cladding mode and the fiber cladding diameter. The solid black line corresponds to the non-etched fiber; the cladding diameter was etched by the 1% hydrofluoric acid solution. Figure 4b shows the spectral evolution of an LPFG with two resonance bands during the etching process. The transmission spectra of the LPFG with two resonance bands could be obtained before the etching process (d_clad_ = 125 µm) and during the etching with closer bands (d_clad_ = 124.6 µm and 124.2 µm). Further reducing the cladding diameter to 123.8 µm, the dual resonance bands were converted into a single, broader resonance band (i.e., up to the DTP). It then vanished if the cladding diameter was continually reduced. In the evaluation of biosensor performance, the Eudragit L100 copolymer was adopted as the chemically modified layer to provide free carboxyl functional groups to immobilize IgG. The obtained LPFG-based sensor was carried out in human serum, where the detection of anti-IgG concentrations as low as 70 μg/L (460 pM) was measured.

Other than tuning the LPFG to a specific higher-order cladding mode at the DTP by reducing the cladding diameter, the modulation of functional layer thickness has also been adopted. The group of Korposh [43] proposed an LPFG-based biosensor using biotin as a bioreceptor for streptavidin (SV) detection. As shown in Figure 5, the silica core–gold shell nanoparticles (SiO_2_:Au NPs) were coated onto fibers using the layer-by-layer method with the aid of a poly (allylamine hydrochloride) (PAH) polycation layer. It was convenient to tune the LPFG-based biosensor operating at the DTP using the layer-by-layer deposition method. The diameter of SiO_2_ NPs was also controlled to study the effect on sensitivity, and the result indicates that larger SiO_2_ NPs (i.e., 300 nm) exhibited higher sensitivity. It could contribute to the more efficient evanescent wave in 300 nm diameter SiO_2_ NPs. Furthermore, the larger SiO_2_ NPs were more conducive to biotin depositing, thus enhancing the ability of adsorption of SV. Finally, they achieved the detection of SV with the lowest measured concentration of 2.5 nM, and this proposed sensor could be applied to targeting clinically relevant protein compounds, only needing to replace the ligand.

On the other hand, it has been proved that a lower-order cladding mode can occur near the DTP by reducing the cladding diameter, and then a more sensitive ability is obtained because of the enhancement of the evanescent field [44]. Enlightened by this, recently, there is an interesting proposal to design a lowest-order cladding mode (LP_0,2_ cladding mode) of an LPFG near the DTP, to obtain a high-sensitivity device [47]. An LPFG near the DTP was fabricated, followed by etching the cladding diameter up to 20 μm, until the appearance of the LP_0,2_ mode near the DTP. The proposed device was integrated in a closed flow cell within an SRI range of 1.333 to 1.3335 RIU for testing, leading to a sensitivity of 8751 nm/SRIU. Although it exhibited an excellent performance as a refractometer, many relevant factors should be considered in actual biosensing. On the basis of the foregoing, the same group developed the above device by using the IgG/anti-IgG as a bioconjugate pair for a biosensing application, achieving a limit of detection (LOD) of 0.16 ng/mL (1.06 pM) [48]. In a word, LPFG-based sensors designed at or near their DTP have been widely used in the detection of immunoglobulin [42,46,64], bacteria [16,45,65,66], DNA [67,68,69] and other targets [70,71,72,73,74].

### 3.2. Mode Transition Effect

As mentioned in the introduction, biochemical sensing is realized by sensing the SRI changes when the bioreceptor layers deposited onto the grating region interact with the target. This principle is essentially due to the fact that a small portion of the cladding mode field, the evanescent field, propagates to the outside of the fiber and interacts with the external environment, resulting in $n_{eff,clad}^{\left(m\right)}$ changes, which depend on the thickness of the interaction region and the penetration depth of the evanescent field [75].

The mode transition (MT) effect has been proposed to optimize the sensitivity of LPFG-based sensors to SRI changes. It can occur by coating the fiber cladding surface with a proper thickness of high RI (HRI) materials. Del Villar et al. have made a comprehensive theory and numerical method in their literature [76]. It demonstrated that HRI coatings could modify one of the cladding modes, resulting in the transition between the lower-order cladding-guided modes (i.e., with a higher effective refractive index) to coating-guided modes, and, henceforward, changes the values of $n_{eff,clad}^{\left(m\right)}$, which could be exploited to enhance the sensitivity of sensors. The same year, Andrea Cusano et al. experimentally confirmed a redistribution of cladding modes by uniformly depositing nanoscale HRI coatings along the LPFG and their effect on RI sensitivity [77].

Enlightened by this, Yang et al. [49] fabricated an LPFG-based methane sensor, working in MT with HRI overlay deposition of polycarbonate (PC)/cryptophane A. They used an automatic dip-coating technique to tune the working point of the sensor in the MT region (bulk RI sensitivity was 3.56 × 10^3^ nm/RIU). The detection of methane was performed with a high sensitivity of 2.5 nm/% and LOD of 0.2 % (*v*/*v*). The group of Esposito reported a single-ended LPFG-based sensor for the detection of butane gas, modified with a nanosized HRI overlay of atactic polystyrene [50]. By means of optimizing the layer thickness range, the working point of the device could be tuned within the MT region. The detection of butane vapor was performed with concentrations up to 1.0 vol%, resulting in 2.2 nm/vol% sensitivity, and having the ability to detect concentrations as low as one-tenth of butane lower than the explosive limit. The same research group also fabricated a multilayer-structure LPFG-based sensor; the structure diagram is shown in Figure 6a [51]. The layer consists of PC film and a much thinner layer of graphene oxide (GO), where the nanosized thickness of the PC film was flexibly controlled using the dip-coating technique, in favor of tuning the device to work in the MT region. Figure 6b shows the wavelength shift of the attenuation band as a function of PC thickness, operating air and PBS solution as surroundings. The second layer of GO was exploited due to its properties of biocompatibility and abundant functional groups, which gave the device the ability to bond biometric molecules; then, the high-affinity streptavidin–biotin system was chosen to evaluate the performance of the device through the detection of biotinylated BSA, achieving a LOD of 0.2 aM.

In addition to using HRI organic materials as coatings, many inorganic materials have also been selected to tune the working point in the MT region. The main advantages of inorganic coating materials are that there is a greater available range of RI values and more growth techniques and that it is easier to obtain uniform thickness compared to organic materials. Piestrzyńska et al. reported a label-free biosensor based on LPFGs, deposited with a thin tantalum oxide (TaO_x_) overlay for RI sensitivity. Since the RI of TaO_x_ was as high as 2 in the IR spectral range, the thickness of overlay was precisely controlled at subnanometer, using atomic layer deposition technique. The RI sensitivity of obtained LPFG-based sensors was 11,500 nm/RIU in the RI range of 1.335 to 1.345 RIU [52]. Continually, the group of Saha theoretically studied the phenomenon of cladding MT by coating with an HRI layer of Si_3_N_4_; the result showed that the RI sensitivity was more than 100 μm/RIU for aqueous solution (i.e., RI= 1.33) [53]. Another recent contribution from the group of Li., who used an LPFG-based sensor coated with Au-Si thin films for RI sensing, was the achievement of an ultrahigh 7267.7 nm/RIU sensitivity (i.e., around RI= 1.315); they attributed this excellent performance to the MT of the EH cladding modes and a strong evanescent field penetrating the surroundings [54].

Although various types of coating materials have been exploited, ranging from organic to inorganic, they still have a negative effect on the obtained device, including its repeatability and long-term stability and absorption loss [55]. Therefore, a new strategy has been employed through using double-cladding fiber (DCF) with a W-shaped RI profile. The group of Esposito realized the MT effect for RI sensitivity by writing the LPFG fabricated in a W-shaped DCF for the first time. The outer cladding RI of the DCF was higher than the inner cladding one; hence, the outer cladding of the DCF acted similarly to the HRI overlay. In order to tune the working point to the MT region of this advice, the outer cladding was etched by a chemical reagent. Finally, attention was focused on the SRI sensitivity, achieving a 420 nm/RIU sensitivity in water-like surroundings, which confirmed that these device could be used for biochemical sensing without an HRI coating [55]. Subsequently, they developed and tested this special structure device, through coating with a nanoscale GO layer to offer functional groups to covalently bond the antibody. A working range from 1 ng/mL to 100 μg/mL and a LOD of 0.15 ng/mL were achieved for the detection of C-reactive protein in serum [56]. Recently, the same group reported a similar DCF biosensor for the detection of vitamin D. As shown in Figure 7, the nanometric GO was also selected to provide carboxylic functional groups for the covalent immobilization of Anti-VitD3. The selective detection with a range of 1–1000 ng/mL in buffer solution was obtained, and these devices also performed well in a complex medium with interfering proteins [57].

### 3.3. The Combination of These Approaches

As mentioned above, the LPFG working at or near the DTP offers a high SRI sensitivity, but the maximum sensitivity is limited to the vicinity of the DTP, indicating that it is difficult for LPFG-based device to maintain the maximum sensitivity in a broad RI range [59]. When the LPFG devices are applied to biochemical sensing, it faces a varying RI range. For example, for the conventional bioreceptor (e.g., protein, DNA, antibody) to detect a biotarget (e.g., protein, antigen), the SRI varies from 1.333 to 1.353 RIU [12,45]. However, for the LPFG modified with gelatin as a humidity sensor, the SRI varies from 1.3408 RIU to the RI of the cladding [78].

Similarly, the MT effect can guide one of the cladding modes to coating-guided modes, and a specified sensitivity range can be tuned by the precise adjustment of the thickness and reducing the cladding diameter [59,61]. It indicates that the combination of both the DTP and MT effect can improve LPFGs’ sensing properties, including the high sensitivity and the specified SRI working range. For example, the group of Mateusz realized an RI sensitivity of 2000 nm/RIU in a broader RI range (1.34–1.356 RIU) [59]. In this work, the cladding diameter and coating thickness were controlled in a nanometer range to reach both the DTP and MT effects, using the method of reactive ion etching (RIE) and radio frequency plasma-enhanced chemical vapor deposition. In addition, many reported works demonstrated that the sensitivity could also be improved by reducing the diameter of the LPFG-based sensor [42,79]. Enlightened by this, the cladding diameter was combined with the DTP and the MT effect for optimizing the sensitivity to the SRI by Del Villar in [60]. A numerical method based on the exact calculation of the core and cladding modes and coupled mode theory was used to analyze the optimization of an LPFG based on the above three factors. Eventually, a considerable sensitivity of 143 × 10^3^ nm/RIU was obtained, which was expected to improve the resolution of the LPFG-based chemical and biological sensors. Recently, Fang et al. also adequately combined both the DTP and MT effects using TiO_2_ as the high RI nanofilm deposited by atomic layer deposition technology [61]. To keep the spectral difference of the dual peaks constant, the appropriate grating period was selected in different LPFGs, and different TiO_2_ thicknesses were investigated to optimize the sensitivity in a specific SRI range. As shown in Figure 8, it showed an SRI sensitivity of 10,000 nm/RIU in the range 1.336–1.3397 RIU, 42,000 nm/RIU in the range of 1.4526–1.4561 RIU, 15,000 nm/RIU in the range of 1.392–1.3971 RIU and 23,000 nm/RIU in the range of 1.44–1.4436 RIU. From Figure 8c,d, it indicates that two high-sensitivity ranges could be realized in the same LPFG device.

## 4. The Functionalization of LPFG-Based Biosensors

It is important to note that the LPFG-based biosensor is applied to measure the wavelength shift caused by the RI change of the device surface due to the selective adsorption for target molecules on the surface, rather than measuring the wavelength shift caused by the change of the RI of the bulk surrounding medium. Therefore, the functionalization of the LPFG-based biosensor is a fundamental step in realizing biochemical applications. Generally, two-part functional layers are deposited onto the surface of the LPFG-based biosensor: one part is the biocarrier layer used as the immobilization of the bioreceptor, and the other is the bioreceptor layer used as the recognition element (enzymes, proteins, antibodies, and so on) to selectively capture the target. Therefore, various methods have been employed for the immobilization of the bioreceptor layer onto the optical fiber. The comparison between the bioreceptor, target and performance of the different functionalization methods of LPFG-based biosensors can be seen in Table 2.

### 4.1. APTES Silanization

The most effective method for LPFG-based biosensor functionalization is based on covalent immobilization, due to its permanent attachment to the bioreceptor. The 3-aminopropyltriethoxysilane (APTES) silanization is a common and ideal method used in most chemical modifications of silica substrates. This method was successfully realized for the covalent immobilization of protein [101,102], DNA [67,68], antibodies [17,82,103] and so on.

In this case, the optical fiber requires a pretreatment step to form silanol groups (Si-OH), by immersing in KOH/NaOH, acid, or piranha solution. Similarly, ethoxy (–OCH_2_CH_3_) groups existing in the APTES molecule can also form the Si-OH through a hydrolysis reaction in aqueous environments [104]. Then, the condensation between Si-OH leads to the formation of a siloxane (Si-O-Si) bond, allowing the APTES molecules to immobilize onto the fiber surface. In addition, the adjacent APTES molecules can form a polymer matrix through condensation, resulting in the formation of free amino-functional (–NH_2_) surfaces of silica substrates [105]. After the silanization is accomplished, there is a step of activation of thecarboxyl groups on antibodies or enzymes, with the aid of 1-ethyl-3- (3-dimethylaminopropyl) carbodiimide hydrochloride (EDC) and N-hydroxysuccinimimide (NHS). Then, the antibodies or enzymes can be bonded to the –NH_2_ groups via forming amide bonds, hydrogen bonding, or electrostatic interaction [106].

The group of Anjli proposed an enzymatic biosensor based on LPFG through the stable covalent binding of the lipase enzyme for the detection of triacylglycerides [80]. Figure 9 shows the step of the immobilization of the lipase enzyme; the lipase enzyme was bonded to the fiber surface by forming amide bonds between –NH_2_ groups on the fiber and –COOH groups of the enzyme. The wavelength shift was measured in association with the enzyme interacting with the target. The detection of triacylglyceride concentrations as low as 17.71 mg/dL was measured; a specific test was also carried out in human blood, and the entire experiment was operated at a constant temperature of 37 °C. APTES silanization to immobilize glucose oxidase was also performed by Wu in [81]. In this work, glucose oxidase was immobilized onto the S-shaped LPFG by APTES silanization technology and used as the bioreceptor for the detection of glucose. The transmission loss variation was used as a measurement associated with glucose oxidase and glucose-specific binding. The experimental results show that the proposed sensor performed a sensitivity of 6.229 dB/wt% in a range from 0∼1 wt%. More recently, Gan et al. [39] developed an LPFG-based sensor based on egg yolk antibody (IgY) covalently immobilized by APTES silanization for the detection of *Staphylococcus aureus*. The detection test could be completed in about 20 min, and the detection of *Staphylococcus aureus* was performed down to 33 CFU/mL. Therefore, the developed device for the detection of *Staphylococcus aureus* was expected to be applied to the field of medical and food detection.

### 4.2. GO Functionalization

Although the amino functionalized fiber is widely proposed, the single functional group makes it unable to bond to other kinds of biological receptors, thus limiting its application. Different research groups have focused their attention on GO; the fiber surface is chemically or physically deposited with GO nanosheets after silanization. The GO is rich in oxygen-containing functional groups, such as epoxy, hydroxyl and carboxyl, which provide GO the capability to covalently bond various biomolecules [107]. Moreover, the GO is also endowed with the capability to adsorb biomolecules by noncovalent immobilizing, such as electrostatic, interaction hydrogen bonding and π–π stacking [108,109].

Chen et al. [46] reported a dual-peak LPFG deposited with GO nanosheets to immobilize the IgG and the IgG/anti-IgG as a bioconjugate pair for immunosensing. They adopted a new strategy to deposit GO that relied on chemical bonding followed by physical adsorption. Chemical bonding occurred between the APTES-silanized fiber and GO. In the meanwhile, GO nanosheets were physically adsorbed on the fiber surface, along with the evaporation of water. Finally, the GO-deposited LPFG-based sensor was immersed into IgG solution and covalently bonded together via EDC/NHS cross-linking chemistry. The detection of anti-IgG was obtained down to a concentration of 7 ng/mL in PBS buffer. The reusability of the sensor was also carried out by stripping off bound anti-IgG. Successively, the same group developed the detection of hemoglobin, based on a GO nanosheet-functionalized LPFG-based sensor [85]. As shown in Figure 10, the chemical bonding followed by physical adsorption strategy was also adopted for depositing GO, and the sensing principle was based on the measurement of variation in the resonance intensity, caused by noncovalent interaction between the hemoglobin molecules and GO. The desirable GO thickness was controlled to 501.8 nm, which provided a significant light–matter interaction between evanescent field and target molecules. The developed sensor performed with a sensitivity of −77 dB/RIU and a LOD of 0.05 mg/mL for hemoglobin detection.

Recently, the detection limits and deposition method have been further improved by Wang et al. in [86]. In this work, a micro-tapered LPFG was deposited with GO nanosheets for the detection of hemoglobin. After completing the chemical bonding followed by physical adsorption, they took advantage of the optical tweezer effect to further enhance the interaction between GO nanosheets and the fiber. The GO thickness of 203.6 nm was immobilized onto the micro-tapered LPFG. The LOD of 0.02 mg/mL in various interfering compounds was obtained. More recently, the same group developed a biosensor with the same structure for bovine serum albumin (BSA) detection [87]. The sensing mechanism relies on measuring the wavelength shift caused by the covalent bonding between the GO and BSA. The LOD of 0.043 mg/mL, 0.029 mg/mL and 0.032 mg/mL were achieved in DI water, urea and glucose, respectively. Similarly, they also proposed a micro-tapered LPFG functionalized by GO/polydopamine nanocomposites for cobalt ion sensing [88]; the nanocomposites were deposited onto the micro-tapered LPFG surface on account of APTES silanization followed by the optical tweezers effect. The proposed sensor showed a sensitivity of 2.4 × 10^−3^ dB/ppb in the cobalt ion concentration range from 1 ppb to 10^7^ ppb, and a detection limit of concentrations as low as 0.17 ppb was achieved. The micro-tapered LPFG-based sensors functionalized by GO have also been implemented for Na^+^ and Mn^2+^ ion detection in [110] and Ni^2+^ ion detection in [89].

### 4.3. Layer-by-Layer Assembly Method

The method of covalent modification of fibers introduced above has good stability; however, it is more complicated to control the thickness, such as controlling the reaction concentration and time. The realization of functional coating with a controllable thickness is also a factor to be considered when optimizing the sensing sensitivity of the LPFG-based device. The layer-by-layer (LbL) assembly provides a promising way to precisely deposit a functional coating with a nanometer-scale thickness [70,111], driven by electrostatic interactions between oppositely charged polyelectrolytes [112,113].

The group of Tian [90] developed a salinity sensor based on LPFG coated with ionic-strength responsiveness of chitosan (CHI)/poly (acrylic acid) (PAA) polyelectrolyte multilayers by the LbL assembly method. The entire deposition process was repeated 20 times by immersion of the device into a polycation CHI and polyanion PAA, respectively. Interestingly, the LPFG resonance wavelength shift changed from red to blue with increasing salt concentration. It could be explained by the de-swelling or swelling of the coating in response to a different range of NaCl concentration. The sensitivity of 36 nm/M was obtained in the range of 0.5–0.8 M. This research was also expected to apply to biomedicine and drug delivery. Similarly, the same group also proposed a salinity sensor which coated with ionic strength-responsive hydrogel onto the LPFG [91]. The two-component polyelectrolytes deposited by LbL assembly might cause some problems, i.e., pH cross-sensitivity and nonlinear relations between the resonance wavelength shift and the concentration of salinity. As shown in Figure 11, this work fabricated a novel sensor coated with a quaternized poly (4-vinylpyridine) (qP4VP) hydrogel via LbL assembly, followed by chemical cross-linking. It is worth noting that the polyanion PAA was selectively released from the coating after chemical cross-linking to obtain a single-component hydrogel. The developed sensor exhibited a sensitivity of 7 nm/M, and there was a good linear relationship between the resonance wavelength shift and salinity concentration in the range of 0.4–0.8 M.

Other polyelectrolyte functional coatings were also deposited onto LPFG-based biosensor by the LbL assembly method. An LPFG-based biosensor coated with nano-assembled thin film of poly (diallyldimethyammonium chloride) (PDDA) and tetrakis (4-sulfophenyl)porphine (TSPP) via the LbL technique for ammonia gas detection was fabricated by Lee et al. [92]. The group of Abd-Rahman fabricated a PDDA/poly (sodium-p-styrenesulfonate) (PSS)-Au nanoparticle coating layer onto an LPFG surface by using the LbL technique for mercury (II) ion sensing [93]. The designed sensor had an excellent performance in the mercury (II) ion concentration range of 0.5 ppm to 10 ppm. Liu et al. [64] developed an LPFG-based biosensor coated with poly (allylamine hydrochloride) (PAH)/gold-coated silica nanoparticles via the LbL method for the detection of streptavidin and immunoglobulin M (IgM). Ni et al. [94] investigated an LPFG-based sensor with a coating of poly (ethylenimine) (PEI) and poly (acrylic acid) (PAA) for pH sensing; the coating layer improved the dispersion and the adhesion ability of multi-walled carbon nanotubes. The group of Tian [95] considered PAH/PAA as a polyelectrolyte functional coating deposited by LbL assembly to bond specific antibodies for the detection of *Staphylococcus aureus.* The polyelectrolyte functional coating could facilitate the bacterial adhesion, and the detection with a LOD of 224 CFU/mL was demonstrated in PBS.

### 4.4. Other Methods

The method of optical fiber functionalization depends on the application environment to a certain extent. The functionalization method of optical fibers applied in air is more concise and simpler than those applied in aqueous solution. For example, the dip-coating technique, which is simple to operate and makes it easy to control the thickness of sensitive film, is widely used in gas sensing.

The group of Feng [96] reported the molybdenum sulfide/citric acid composite films that were deposited by using the sol–gel and dip-coating techniques onto an LPFG for measuring trace hydrogen sulfide gas. Figure 12 shows the SEM images of the surface and cross section of the LPFG coated with molybdenum sulfide/citric acid composite films. It can be observed from Figure 12b that the thickness of the film was uniformly controlled at 590 nm. The proposed sensor exhibited a high sensitivity of 10.52 pm/ppm within a hydrogen sulfide gas concentration range from 0 to 70 ppm, and the hydrogen sulfide gas was detected down to a concentration of 0.5 ppm. The same group also studied the tapered LPFG-based sensor functionalized by molybdenum sulfide/citric acid composite films for hydrogen sulfide gas sensing [114]. A sensitivity of 16.65 pm/ppm was achieved. In addition, an LPFG-based sensor coated with GO was developed for nitric oxide (NO) gas detection by the group of Ding [97]. The coatings were prepared by using the dip-coating technique after a 5% HNO_3_ solution treatment. The NO sensor exhibited excellent sensing features in the NO concentration range of 0 to 400 ppm. Xu et al. proposed an LPFG-based sensor deposited with GO/cellulose acetate composites for ammonia sensing [115]. The chemical cross-linking method and dip-coating technique were taken to deposit these composites. The proposed ammonia sensor exhibited excellent sensitivity (98.32 pm/ppm).

In addition, the metal organic frameworks (MOFs), because of their excellent properties of tunable porosity, large internal surface area and organic functionality, have been widely applied for functionalizing LPFG-based sensors for gas sensing and other sensing. MOFs are hybrid crystalline nanomaterials composed of metal cations and organic ligands [116,117,118]. The methods of functionalizing optical fibers with MOFs mainly focus on in situ crystallization [119,120]. The group of Korposh developed an LPFG-based organic vapor sensor functionalized by zeolitic imidazole framework-8 (ZIF-8) films [121]. As shown in Figure 13a, the ZIF-8 films were deposited onto the LPFG surface using the in situ crystallization technique. The transmission spectrum of the LPFG was monitored during each modification step. The uniform thickness of the ZIF-8 is shown by the SEM images in Figure 13b. Finally, this proposed sensor was performed with a sensitivity of 0.015 ± 0.001 and 0.018 ± 0.0015 nm/ppm and a LOD of 6.67 and 5.56 ppm for acetone and ethanol, respectively, in [98]. The same group fabricated an LPFG-based carbon dioxide (CO_2_) sensor modified with HKUST-1 thin film using in situ crystallization and the LbL technique [99]. The detection of CO_2_ was performed with a LOD of 401 ppm. Moreover, the MOFs can also be used as potential matrices for enzymes’ integration [122]. Zhu et al. reported that the glucose oxidase (GOx)-encapsulated ZIF-8 was coated onto the LPFG via in situ crystallization [100]. A sensitivity of 0.5 nm/mM in the range of 1–8 mM for the detection of glucose was obtained.

## 5. Reflective LPFG-Based Sensors

Since an LPFG-based sensor can couple light from the fundamental core mode to forward-propagating cladding modes, it therefore produces a set of resonant attenuation bands centered at discrete wavelengths in the transmission spectrum. One of the peculiarities of LPFG-based sensors is their operation in transmission mode, which is inconvenient for some biological applications when they need to be introduced in vial/test tubes. From the point of view of realizing miniaturization, the light emission port and the signal collection port need to be deployed on the same facet. Moreover, bending sensitivity can bring interference of the transmitted optical signal [123].

Therefore, there are some works where the transmission mode operation of the LPFG is transformed in reflection mode operation. Swart, P. L. proposed a single-probe Michelson interferometer based on an LPFG where the interferometer phase shift depends on the RI of the surroundings [124]. Kim D. W. also developed a single-fiber probe based on two interferometers, the reflection-mode LPFG for RI sensing and an intrinsic Fabry–Perot interferometer for temperature measurement [125]. Although the LPFG-based interferometer sensor operated in reflection mode, the total length of the sensor might be at least 4~5 cm due to the nature of the interferometer [124,126], which was inconvenient for in vivo detection. Jiang et al. developed a compact reflective LPFG sensor by coating only the cladding end facet with an aluminum film, and then reflecting only the cladding modes [127]. The length of the proposed device was normally 0.5~3 cm long, which was the same as the conventional LPFG-based sensor. However, the complex coating process was utilized. Rana S. et al. used a cost-effective brush coating method using available silver paste to realize a reflective LPFG sensor. The developed device was performed with a temperature sensitivity coefficient of 0.046 nm/°C in a range between 23 °C and 200 °C [128].

In [129], a reflective LPFG-based sensor with a Sagnac fiber loop mirror (SFLM) for measuring the RI and temperature was fabricated. The simultaneous RI and temperature measurement could be achieved by this device, due to the different sensitivities to RI and temperature of the LPFG and SFLM. Another unavoidable problem of reflective LPFG-based sensors is the generation of undesired interferometric bands that overlap the LPFG attenuation bands [130,131]. Some effective approaches focus on precisely cleaving at the end of the LPFG [123] or polishing after cleaving [132] to obtain a unique attenuation band. In [123], as shown in Figure 14, the end of the grating was precisely cleaved and coated with a reflecting layer. In order to enhance the SRI sensitivity of this device, the HRI overlay of the atactic polystyrene was modified on the surface of the fiber. Then, the poly (methylmethacrylate)-co-methacrylic acid was coated onto the fiber as a bio-functional layer to covalently immobilize the bioreceptor. The detection of class C β-lactamases was obtained down to a concentration of the order of a few tens of nM in PBS buffer.

Villar I. D. et al. [133] proposed a simpler way to obtain a single attenuation band; the end of the fiber was coated with a silver mirror which could absorb the power transmitted through the cladding modes. More interestingly, Dey T. K. et al. [101] removed a portion of the LPFG at an arbitrary location, without the requirement of precise cleaving or polishing. The unwanted resonance bands could be removed by tailoring the PMC of the cladding mode; it also benefited to enhance the RI sensitivity of this device. Finally, an RI sensitivity of ∼1300 nm/RIU was obtained.

## 6. Conclusions

This review highlights the basic operating principle of LPFG-based sensors and their sensitization and sensitivity enhancement mechanisms for chemical and biomedical applications. The basic principle of LPFG-based biosensors is that the SRI changes can be converted into the measurement of the resonance wavelength shift or transmission loss variation. To enhance the sensitivity of LPFG-based biosensors for chemical and biomedical applications, two main methodologies have been adopted in recent published reports. One is to design an LPFG working at or near a DTP, which can be realized by tuning the grating period, cladding diameter or the functional layer thickness. This methodology has a good performance for the detection of immunoglobulin, bacteria, DNA and other targets. The other one is the MT effect, which can be realized by coating the fiber cladding surface with a proper thickness of HRI materials. This methodology has been proven to perform well in detecting organic gas (e.g., methane and butane) and protein (e.g., biotinylated BSA, C-reactive protein). The combination of these strategies is also reviewed in this paper. On the other hand, the functionalization of LPFG-based biosensors is an indispensable step to achieve specific sensing. As shown in Table 2, we summarize several common functionalization methods corresponding to different kinds of bioreceptors. The APTES silanization is usually applied to covalently bond the antibodies or enzymes, further functionalization with GO can offer device the capability to covalently bond more kinds of bioreceptors due to the abundant functional groups of GO. In addition, LbL assembly, the method of noncovalent functionalization, is also reviewed in this paper. The advantage of this method is that it is more convenient and can precisely control the thickness of the functional layer. This method has been applied to deposit polyelectrolytes for salinity and pH sensing, as well as ion, gas and bacteria detection. Other functionalization methods (e.g., dip-coating and in situ crystallization techniques) have also been reviewed. Moreover, reflective LPFG-based sensors have also been introduced. Although the demonstrated LPFGs in reflection configurations exhibit less sensitivity, they can be introduced in the vial/test tube, which is more convenient and practical for their chemical or biomedical applications.

In this review, it has been shown that the LPFG is a promising sensing platform for chemical and biomedical applications. A meaningful future direction should be exploiting innovative coating materials and functionalization methods to detect more kinds of biomolecules. In addition, the inherent multiplexing capability of OFBS should be utilized in LPFG-based sensors to achieve multi-parameter or multi-target measurements by writing different gratings in the same device. In complex measurement environments, more efforts should be focused on reducing or eliminating the cross-sensitivity (e.g., temperature, strain) effects on LPFG-based sensors.

## Figures and Tables

**Figure 1 sensors-23-00542-f001:**
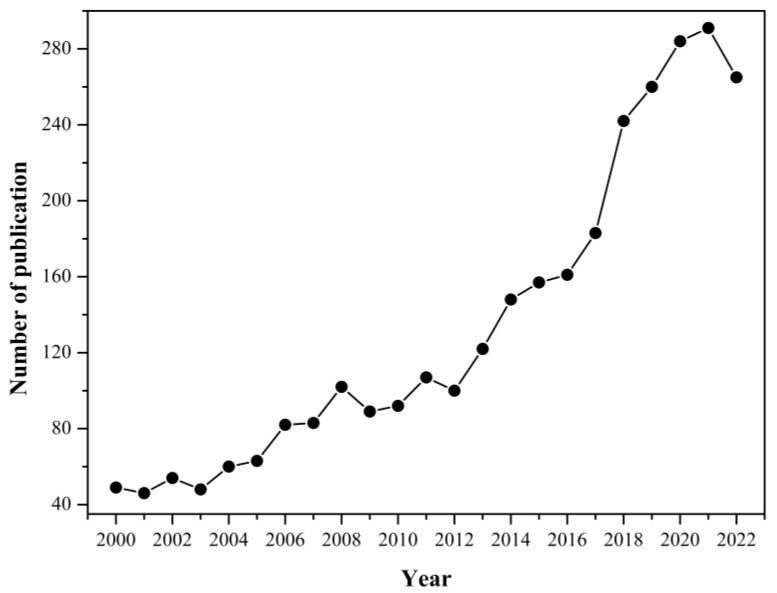
The growth of publications as a function of years for OFBS. Data from Web of Science.

**Figure 2 sensors-23-00542-f002:**
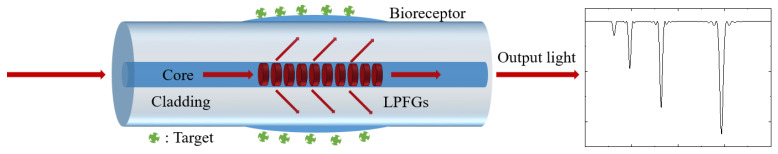
Schematic illustration of OFBS based on an LPFG.

**Figure 3 sensors-23-00542-f003:**
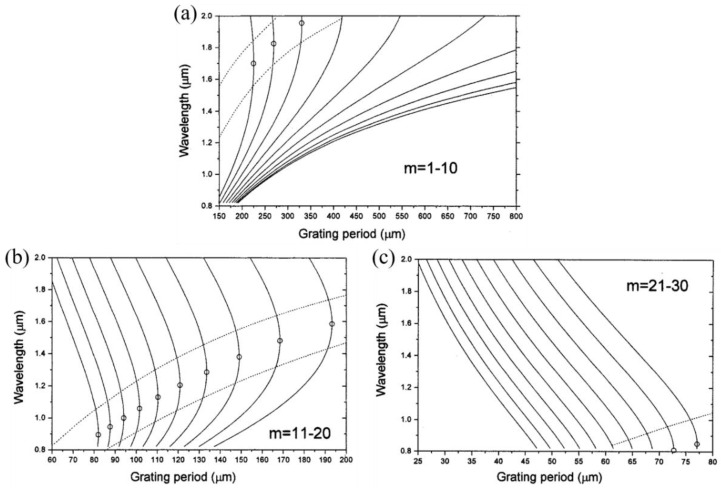
Calculated variation of mode resonance wavelength with LPFG period. (**a**) Modes m = 1 to m = 10. (**b**) Modes m = 11 to m = 20. (**c**) Modes m = 21 to m = 30. The small circles locate the turning points of the slopes of the curves, and the LPFG exhibits greatest sensitivity in the region between the two dotted lines. Adopted with from [18]. Under a Creative Commons license.

**Figure 4 sensors-23-00542-f004:**
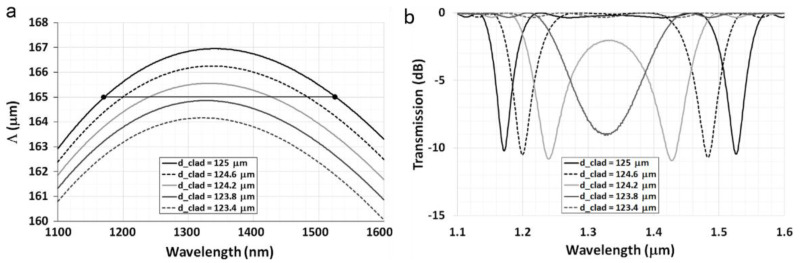
(**a**) PMC for LP_0,12_ cladding mode as a function of the fiber diameter with a grating period of 165 mm (solid black line corresponds to the non-etched fiber, whereas solid dark gray line corresponds to the closest curve to the DTP condition). (**b**) Simulated spectra of LPFG with grating period of 165 µm for decreasing thickness of the fiber diameter d_clad_ from 125 µm up to 123.4 µm, detailing the evolution of the dual resonance bands up to DTP (solid dark grey line). Reprinted with permission from [42]. Copyright © 2014 Elsevier B.V.

**Figure 5 sensors-23-00542-f005:**
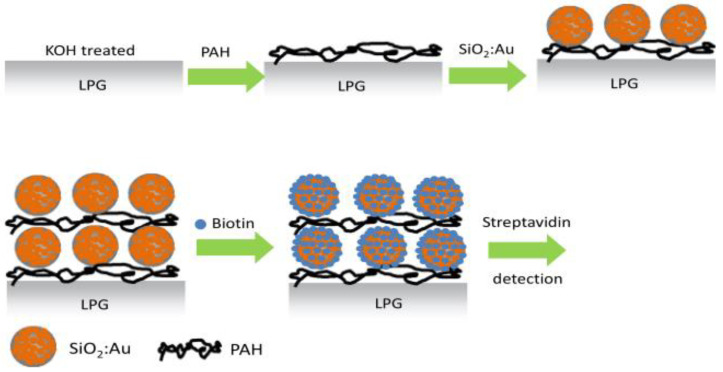
Schematic illustration of the layer-by-layer deposition of a (PAH/SiO_2_:Au)_2_ film onto an LPFG. Adopted from [43]. Under a Creative Commons license.

**Figure 6 sensors-23-00542-f006:**
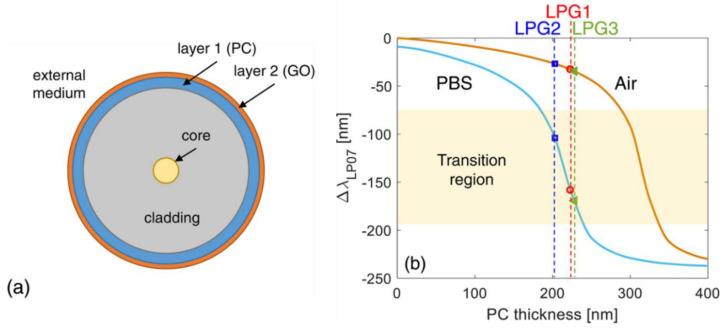
(**a**) The structure of multilayer fibers. (**b**) Numerical wavelength shift of the attenuation band versus PC overlay thickness. Reprinted with permission from [51]. Copyright © 2018 Elsevier B.V.

**Figure 7 sensors-23-00542-f007:**
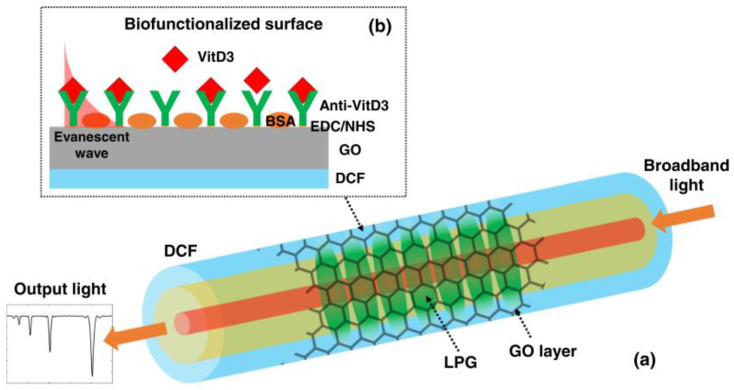
Schematic picture for the detection of vitamin D based on an LPFG in DCF with GO layer. (**a**) Overall schematic picture of the device; (**b**) Details regarding biofunctionalized fiber surface and sensing mechanism (not to scale). Adopted from [57]. Copyright © 2021, Elsevier.

**Figure 8 sensors-23-00542-f008:**
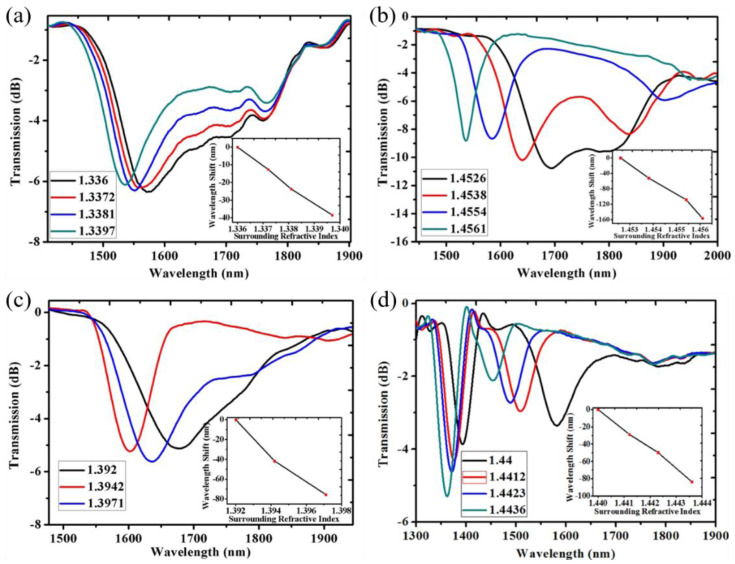
Transmission spectra and resonance wavelength shift in different range of SRI. (**a**) 65 nm TiO_2_ nanofilm, Λ = 230.5 μm. (**b**) 15 nm TiO_2_ nanofilm, Λ = 237 μm. (**c**,**d**) 50 nm TiO_2_ nanofilm, Λ = 230 μm. Adopted from [61]. Under a Creative Commons license.

**Figure 9 sensors-23-00542-f009:**
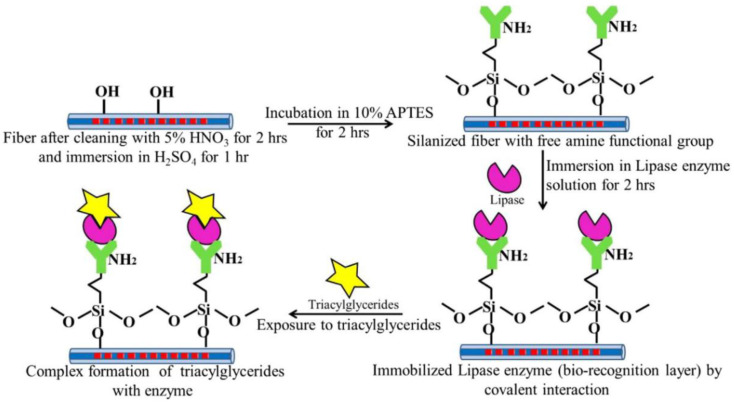
Immobilization of enzymes to create the bio-recognition layer on the optical fiber probe. Reprinted with permission from [80]. Copyright © 2015 Elsevier B.V.

**Figure 10 sensors-23-00542-f010:**
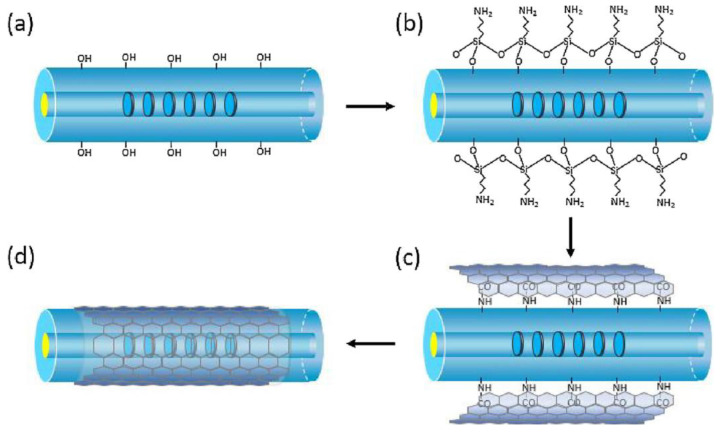
Schematic of GO deposition on LPFG-based device. (**a**) The process of alkaline treatment, (**b**) APTES silanization, (**c**) the epoxy group of GO reacted with amino group of APTES-silanized fiber surface, and (**d**) GO nanosheets were deposited onto fiber surface. Reprinted with permission from [85]. Copyright © 2018 Elsevier B.V.

**Figure 11 sensors-23-00542-f011:**
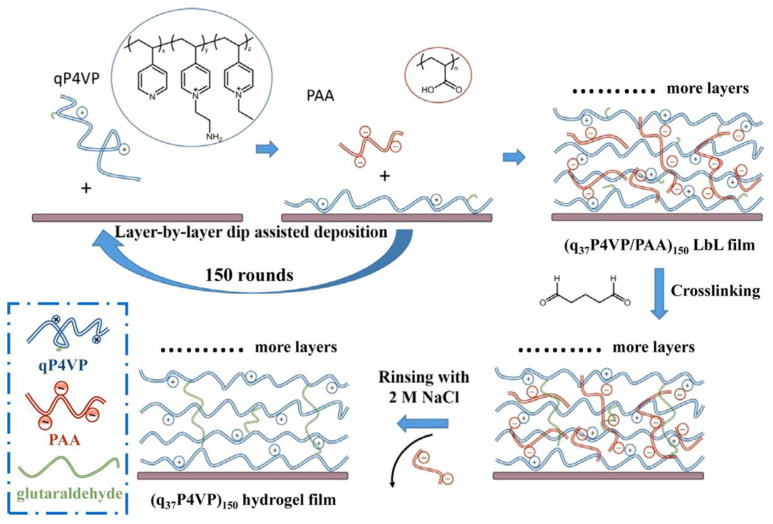
Fabrication of a q_37_P4VP hydrogel coating. Adopted from [91]. Under a Creative Commons license.

**Figure 12 sensors-23-00542-f012:**
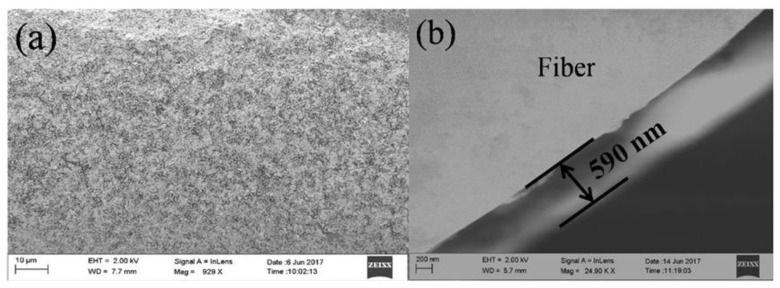
SEM images of (**a**) the side surface of the LPFG and (**b**) cross section of the composite membrane-coated LPFG. Reprinted with permission from [96]. Copyright © 2018 Elsevier B.V.

**Figure 13 sensors-23-00542-f013:**
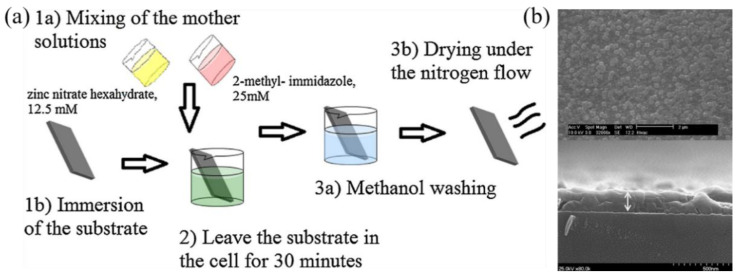
(**a**) ZIF-8 fabrication methodology. (**b**) Top view and cross-sectional SEM images of ZIF-8 films grown on glass substrates. Adopted from [121]. Under a Creative Commons license.

**Figure 14 sensors-23-00542-f014:**
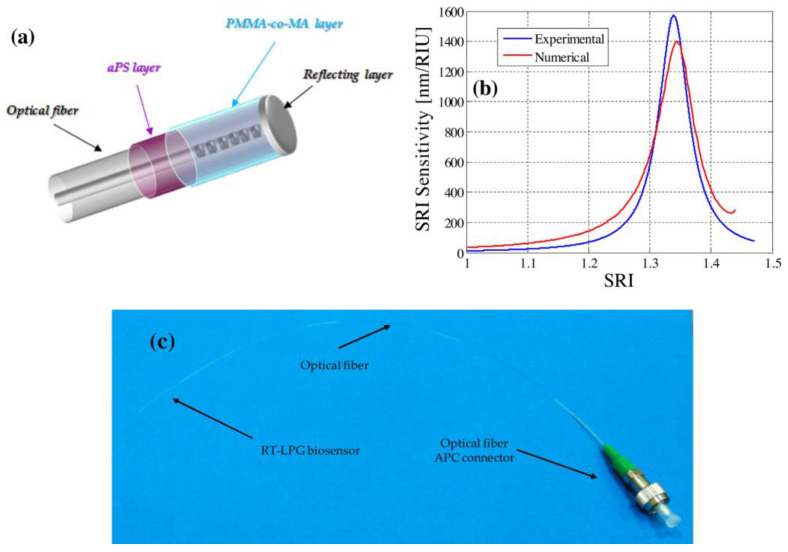
(**a**) Schematic view of the final reflective LPFG transducer; (**b**) the experimental and numerical SRI sensitivity; (**c**) a photograph showing the reflective LPFG biosensor probe developed in this work. Reprinted with permission from [123]. Copyright © 2016 Elsevier B.V.

**Table 1 sensors-23-00542-t001:** Comparison different methods to enhance LPFG-based biosensors.

Enhanced Method	Configuration	Target	Range	LOD or Sensitivity	Ref.
DTP	SM, etched fiber cladding	Anti-IgG	1–100 μg/mL	70 ng/mL	[42]
DTP	B-Ge co-doped fiber	SV	0.3–2.7 μM	2.5 nM	[43]
DTP	SM, etched fiber cladding	-	RI: 1.353–1.413	8734 nm/RIU	[44]
DTP	SM, Λ = 226.8 μm, 358 μm	-	RI: 1.333–1.353	1929 nm/RIU	[45]
DTP	SM, Λ = 162 μm	-	RI: 1.333–1.347	2538 nm/RIU	[46]
DTP	SM, thin cladding fiber	-	-	4298.2 nm/RIU	[32]
DTP	SM, etched fiber cladding.	-	RI: 1.3333–1.3399	7200 nm/RIU	[16]
DTP	Etched fiber cladding	-	RI: 1.333–1.3335	8751 nm/RIU	[47]
DTP	SM, etched fiber cladding.	Anti-IgG	0.1–100 μg/mL	0.16 ng/mL	[48]
MT	PC/cryptophane A overlay	-	-	3500 nm/RIU	[49]
MT	Atactic polystyrene overlay	Butane	~1.0 vol%	−2.2 nm/vol%	[50]
MT	PC/GO overlay	BSA	0.1–1000 aM	0.2 aM	[51]
MT	TaO_x_ overlay	-	RI: 1.335–1.345	11,500 nm/RIU	[52]
MT	Si_3_N_4_ overlay	-	RI: around 1.33	10,000 nm/RIU	[53]
MT	Au-Si overlay	-	RI: around 1.315	7267.7 nm/RIU	[54]
MT	DCF	-	RI: water-like	420 nm/RIU	[55]
MT	DCF	C-reactive protein	1–100 μg/mL	0.15 ng/mL	[56]
MT	DCF	vitamin D	1–1000 ng/mL	1 ng/mL	[57]
DTP+MT	TiO_2_ overlay	-	RI: 1.334–1.340	6200 nm/RIU	[58]
DTP+MT	Diamond-like carbon nano overlay	-	RI: 1.3344–1.3355	12,000 nm/RIU	[59]
			RI: 1.340–1.356	2000 nm/RIU	
DTP+MT	Thin film with RI of 1.55	-	RI: 1.330–1.331	143000 nm/RIU	[60]
DTP+MT	TiO_2_ overlay	-	RI: 1.336–1.3397	10,000 nm/RIU	[61]
			RI: 1.392–1.3971	15,000 nm/RIU	
			RI: 1.44–1.4436	23,000 nm/RIU	
			RI: 1.4526–1.4561	42,000 nm/RIU	

Abbreviations: dispersion turning point (DTP), mode transition (MT), single mode (SM), double cladding fiber (DCF).

**Table 2 sensors-23-00542-t002:** Comparison of different reports of the functionalization of LPFG-based biosensors.

Functionalization Method	Bioreceptor	Target	Rang	LOD or Sensitivity	Ref.
APTES silanization	Lipase enzyme	Triacylglycerides	-	0.2 mM	[80]
APTES silanization	Glucose oxidase	Glucose	0∼1 wt%	6.229 dB/wt%	[81]
APTES silanization	IgY	*Staphylococcus aureus*	10^2^–10^7^ CFU/ml	33 CFU/mL	[39]
APTES silanization	T4 bacteriophage	*Escherichia coli*	-	100 CFU/mL	[45]
APTES silanization	Anti-E. coli antibody	*Escherichia coli*	-	7 CFU/mL	[82]
APTES silanization	DNA	DNA	-	4 nM	[68]
APTES silanization	DNA aptamer	*Escherichia coli*	-	10 nM	[83]
APTES silanization	Glucose oxidase	*Aspergillus niger*	-	1000 CFU/mL	[84]
GO functionalization	IgG	Anti-IgG	-	7 ng/mL	[46]
GO functionalization	GO	Hemoglobin	0–1 mg/mL	0.05 mg/mL	[85]
GO functionalization	GO	Hemoglobin	0–2 mg/mL	0.02 mg/mL	[86]
GO functionalization	GO	BSA	0–2 mg/mL	0.043 mg/mL	[87]
GO functionalization	GO/polydopamine	Co^2+^ ions	1 ppb to 10^7^ ppb	0.17 ppb	[88]
GO functionalization	GO	Ni^2+^ ions	1 ppb to 10^7^ ppb	0.27 ppb	[89]
LbL assembly	CHI/PAA	NaCl	0.5–0.8 M	36 nm/M	[90]
LbL assembly	qP4VP	NaCl	0.4–0.8 M	7 nm/M	[91]
LbL assembly	PDDA/TSPP	NH_3_	-	0.67 ppm	[92]
LbL assembly	PDDA/PSS-Au	Hg^2+^	0.5 ppm–10 ppm	0.5 ppm	[93]
LbL assembly	PAH-Au:SiO_2_	Streptavidin	1.25 Nm–2.7 μM	1.25 nM	[64]
LbL assembly	PEI/PAA	pH	pH: 2–13	0.83 dB/pH	[94]
LbL assembly	PAH/PAA	*Staphylococcus aureus*	-	224 CFU/mL	[95]
Dip-coating technique	MoS_2_/citric acid	H_2_S	0–70 ppm	0.5 ppm	[96]
Dip-coating technique	GO	NO	0–400 ppm	63.65 pm/ppm	[97]
In situ crystallization	ZIF-8 films	Acetone	6.67 ppm	0.015 nm/ppm	[98]
		Ethanol	5.56 ppm	0.018 nm/ppm	
In situ crystallization	HKUST-1 film	CO_2_		401 ppm	[99]
In situ crystallization	GOx/ ZIF-8	Glucose	1–8 mM	0.5 nm/mM	[100]

## Data Availability

Not applicable.

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
