# Peer review of "Long-Period Fiber Grating Sensors for Chemical and Biomedical Applications"

_sensors, 2023, doi:10.3390/s23010542_

Round 1

Reviewer 1 Report

This review article is well-written and well organized as well as provides up-to-date information on the current advancement of LPFG-based sensors for chemical and biomedical applications. Hence, I recommend it for publication in its current form.

Author Response

Thank you very much for your positive comment on our manuscript.

Reviewer 2 Report

Dear Editor

This study investigated optical fiber biosensors (OFBS) based on long-period fiber gratings (LPFG) for chemical and biomedical applications. The structure of the manuscript is good.

1.       Abstract should be improved

2.       Introduction needs a curve for the growth of publication as a function of years for optical fiber biosensors

3.       Equation (1) need a reference

4.       Compared different reports of LPFG biosensors in a table (in section 3)

5.       Compared different reports of the functionalization of LPFG biosensors in a table (in section 4)

6.       Quality of Fig.11 need to be improved

7.       Conclusion needs to be supported by data from sections 3 and 4

Reviewer 3 Report

Review article:

Long Period Fiber Grating Sensors for Chemical and Biomedical Applications

The authors present a review about Long Period Fiber Grating Sensors for Chemical and Biomedical Applications.

This is a well-written and structured manuscript. In this manuscript there are the potentialities to publications in Sensors but the paper have need some important specification and reviews.

1- Introduction

In a review it would be useful to have a very broad introduction that takes into account many articles published in this field. Here are some articles that should be included both in the introduction and in the text:

- At line 43 when the authors introduce some examples about the U-shaped fiber, they have to introduce also the LPG evanescent wave in D-shaped fiber (Evanescent wave long-period fiber grating within D-shaped optical fibers for high sensitivity refractive index detection, SnB, 2011  ---- S. Jang, K.N. Park, J.P. Kim, O.J. Kwon, Y.-G. Han, K.S. Lee, Sensitive DNA biosensor based on a long-period grating formed on the side-polished fiber surface, Opt. Express 2009) and also some paper about the LPG in TAP mode.

- At line 45 when the authors introduce the LPG can't insert only 2 paper. The other important paper are: 1- Zuppolini et al. Label-free fiber optic optrode for the detection of class C β-lactamases expressed by drug resistant bacteria in 2017.

- At line 47 the authors have introduce the Lab on Fiber concept in order to realize the nanostructure onto the optical fiber with high precision and expensive cost (Consales at al. Metasurface-Enhanced Lab-on-Fiber Biosensors, 2020) or low cost optrode (Manago et al. Tailoring lab-on-fiber SERS optrodes towards biological targets of different sizes, 2021  -   Pisco at al. Self-assembled periodic patterns on the optical fiber tip by microsphere arrays, 2015).

1- Principle of LPFG-based biosensors

The figures 1 have to be improving about the optical spectrum. The resonance visibility of an LPG have to increasing with respect to the wavelength.

The authors can be see the spectrum in figure 6 of the same paper.

In the paper an interesting section can be the LPG in reflection configuration that is a crucial point in order to use the LPG as biosensors to introduce in the vial/test tube, were there is the biological solution.

Round 2

Reviewer 2 Report

Dear Editor 

The manuscript well has been revised 

Reviewer 3 Report

The manuscript can be published in the revised form.